# Modulatory Effects of Physical Activity Levels on Immune Responses and General Clinical Functions in Adult Patients with Mild to Moderate SARS-CoV-2 Infections—A Protocol for an Observational Prospective Follow-Up Investigation: Fit-COVID-19 Study

**DOI:** 10.3390/ijerph182413249

**Published:** 2021-12-16

**Authors:** Fábio Santos Lira, Telmo Pereira, Luciele Guerra Minuzzi, Caique Figueiredo, Tiago Olean-Oliveira, Ana Paula Coelho Figueira Freire, Manuel João Coelho-e-Silva, Armando Caseiro, Ronaldo Vagner Thomatieli-Santos, Vanessa Ribeiro Dos Santos, Luis Alberto Gobbo, Marília Seelaender, Karsten Krüger, Ricardo Aurino Pinho, José Cesar Rosa-Neto, Bruna Spolador de Alencar Silva

**Affiliations:** 1Exercise and Immunometabolism Research Group, Postgraduation Program in Movement Sciences, Department of Physical Education, Universidade Estadual Paulista (UNESP), Presidente Prudente 19060-900, SP, Brazil; lucielegm@gmail.com (L.G.M.); caiquefigueiredo22@gmail.com (C.F.); tiago.olean94@gmail.com (T.O.-O.); brunaspolador@gmail.com (B.S.d.A.S.); 2Department of Clinical Physiology, Polytechnic Institute of Coimbra, Coimbra Health School, Rua 5 de Outubro-SM Bispo, Apartado 7006, 3046-854 Coimbra, Portugal; telmo@estescoimbra.pt (T.P.); armandocaseiro@estescoimbra.pt (A.C.); 3Laboratory for Applied Health Research (LabinSaúde), Rua 5 de Outubro-SM Bispo, Apartado 7006, 3046-854 Coimbra, Portugal; 4Physiotherapy Department, Universidade do Oeste Paulista (UNOESTE), Presidente Prudente 19050-900, SP, Brazil; anapcff@hotmail.com; 5Centro de Investigação do Desporto e da Atividade Física, Faculdade de Ciências do Desporto e Educação Física, Universidade de Coimbra, CIDAF, 3030-779 Coimbra, Portugal; mjcesilva@hotmail.com; 6Department of Bioscience, Universidade Federal de São Paulo (UNIFESP), Santos 11000-000, SP, Brazil; ronaldo.thomatieli@unifesp.br; 7Skeletal Muscle Assessment Laboratory Postgraduation Program in Movement Sciences, Department of Physical Education, Universidade Estadual Paulista (UNESP), Presidente Prudente 19060-900, SP, Brazil; vanessa.ribeiro-santos@unesp.br (V.R.D.S.); luis.gobbo@unesp.br (L.A.G.); 8Cancer Metabolism Research Group, LIM26-HC, FMUSP, University of São Paulo, São Paulo 11000-000, SP, Brazil; seelaender@usp.br; 9Department of Exercise Physiology and Sports Therapy, Institute of Sports Science, Justus-Liebig-University Giessen, 35390 Giessen, Germany; Karsten.Krueger@sport.uni-giessen.de; 10Graduate Program in Health Sciences, School of Medicine, Pontificia Universidade Catolica Do Parana, Curitiba 80000-000, PR, Brazil; rapinho12@gmail.com; 11Immunometabolism Research Group, Department of Cell and Developmental Biology, Institute of Biomedical Sciences, Universidade de São Paulo (USP), São Paulo 01000-000, SP, Brazil; josecesar23@hotmail.com

**Keywords:** SARS-CoV-2, inflammation, exercise, adipose tissue, immunity, long-COVID

## Abstract

**Background:** This proposal aims to explain some of the gaps in scientific knowledge on the natural history of coronavirus disease (COVID-19), with a specific focus on immune, inflammatory, and metabolic markers, in parallel with temporal assessment of clinical and mental health in patients with COVID-19. The study will explore the temporal modulatory effects of physical activity and body composition on individual trajectories. This approach will provide a better understanding of the survival mechanisms provided by the immunomodulatory role of physical fitness. **Methods:** We will conduct a prospective observational cohort study including adult patients previously infected with the SARS-CoV-2 virus who have expressed a mild to moderate COVID-19 infection. Procedures will be conducted for all participants at baseline, six weeks after vaccination, and again at 12 months. At each visit, a venous blood sample will be collected for immune phenotypic characterization and biochemistry assays (inflammatory and metabolic parameters). Also, body composition, physical activity level, cardiovascular and pulmonary function, peripheral and respiratory muscle strength, functional exercise capacity, and mental health will be evaluated. Using the baseline information, participants will be grouped based on physical activity levels (sedentary versus active), body composition (normal weight versus overweight or obese), and SARS-CoV-2 status (positive versus negative). A sub-study will provide mechanistic evidence using an in-vitro assay based on well-trained individuals and age-matched sedentary controls who are negative for SARS-CoV-2 infection. Whole blood will be stimulated using recombinant human coronavirus to determine the cytokine profile. Peripheral blood mononuclear cells (PBMCs) from healthy well-trained participants will be collected and treated with homologous serum (from the main study; samples collected before and after the vaccine) and recombinant coronavirus (inactive virus). The metabolism of PBMCs will be analyzed using Respirometry (Seahorse). Data will be analyzed using multilevel repeated-measures ANOVA. **Conclusions:** The data generated will help us answer three main questions: (1) Does the innate immune system of physically active individuals respond better to viral infections compared with that of sedentary people? (2) which functional and metabolic mechanisms explain the differences in responses in participants with different physical fitness levels? and (3) do these mechanisms have long-term positive modulatory effects on mental and cardiovascular health? Trial registration number: Brazilian Registry of Clinical Trials: RBR-5dqvkv3. Registered on 21 September 2021.

## 1. Background

A high number of individuals who did not require hospitalization continue to present with sequelae months after contracting mild or moderate Coronavirus disease (COVID-19) [1]. The most common symptoms are fatigue, dyspnea, anosmia, ageusia, headaches, chest and muscle-skeletal pain, exercise intolerance, palpitations, memory loss, and mood changes [1,2]. Therefore, the presence of a “long-COVID” syndrome highlights the need for long-term health follow-up in a subgroup of patients with mild to moderate COVID-19 form who are often overlooked.

The etiology of COVID-19 progression involves interaction between the virus and angiotensin-converting enzyme 2 (ACE2) receptors which are present on cells in nearly every human tissue, suggesting vast consequences for general human physiological functions. Early investigations revealed that severe acute respiratory syndrome coronavirus 2 (SARS-CoV-2) primarily infected the lungs and then spread via the blood circulatory system to different organs and systems, causing dysfunction in musculoskeletal, cardiac, and nervous systems homeostasis [3]. Furthermore, prolonged systemic cytokine-induced inflammatory response and propagation can have deleterious effects downstream of the initial viral infection in the lung, possibly causing severe physiological impairments. On the other hand, physical inactivity (sedentarism) and overweight/obesity can cause greater vulnerability to the immune system and an increased probability of disease progression and subsequent adverse events after infection. However, the effects of physical activity on long-COVID are not well documented.

The benefits of physical activity are well documented and include a reduction in metabolic diseases, cardiovascular diseases, viral infections, and oncological diseases [4]. Regular physical activity is also believed to positively modulate the immune system’s ability to deal with infections. Active individuals seem to have improved immunity when it comes to the recognition and elimination of antigens, as well as the organization of the immune response to protect from or attenuate the symptoms of infection [5]. Studies have linked the positive effect of the immune system and a better pro/anti-inflammatory balance to different intensities of intermittent or regular exercise sessions in subjects who exercise and/or in lifelong athletes compared with sedentary subjects [4,5,6,7,8,9,10,11]. However, we didn’t know if it provides potential protection against SARS-CoV-2 infection and/or can benefit the recovery of people infected with the virus, alleviating the symptoms. Studies of patients with severe COVID-19 have already reported increases in inflammatory monocytes and neutrophils, and a sharp decrease in lymphocytes [12,13], and an inflammatory milieu [14]. Despite these analyses, the dynamics of the immune response during SARS-CoV-2 infection in patients with mild and moderate symptoms and its association with physical activity level and body composition remain unclear.

The level of physical activity has been associated with a positive modulation of general clinical functions, including those associated with the cardiorespiratory, musculoskeletal, and nervous systems [6,8,15,16]. Lifelong physical activity has been associated with better cardiovascular function, indicating the combined effect of a reduction in peripheral vascular resistance, which may be due to neuro-humoral and structural responses, with a reduction in sympathetic nerve activity and an increase in arterial diameter [17]. This promotes better arterial compliance, from which a better buffering of the left ventricle ejection wave, lesser reflected waves, and better heart-arterial coupling are expected [18,19]. Physical activity benefits pulmonary function, mainly due to improvement in the dynamics of the respiratory system, which is promoted by the positive change in body composition and improvement in the resistance and strength of the respiratory muscles. In addition to a more efficient cardiorespiratory system promoting a better supply of oxygen and nutrients to the muscles under constant activity, the metabolic, physical, immunological, and inflammatory changes promoted by physical exercise provide improved functional capacity and quality of life. The mental health benefits of physical activity are also well established [20]. Studies have shown that physical activity is associated with a reduced risk of mental disorders, specifically depression and anxiety [21,22]. Exercise promotes physiological and psychological effects, including increasing levels of endorphins [23], production of neurotransmitters [24], the decreasing response of the hypothalamic-pituitary adrenal axis to stress [24], and promoting positive feelings associated with well-being [25]. Furthermore, it appears that the anti-inflammatory effects of exercise contribute to better mental health outcomes in patients with inflammatory disorders [26]. These are possible mechanisms through which regular physical activity could alleviate the severity of the COVID-19 pandemic, in addition to its positive effects on several chronic diseases [27].

Considering the previous rationale and the current pandemic, it is imperative to ask: What are the long-term consequences of COVID-19 and how do physical activity, body composition, immune profiles, and inflammation contribute to different trajectories of general clinical functions throughout a lifetime and overall physiological recovery? There is no data on the prevalence, nature, and behavior of COVID-19 in physically active individuals, and little is known about the long-term health consequences of COVID-19 and its relation with SARS-CoV-2 vaccination [28].

This proposal describes methods for an observational prospective follow-up investigation of adult patients previously infected with mild to moderate COVID-19, we will conduct measures 3 times, (1) baseline, (2) six weeks after full vaccination, and (3) again at 12 months. Investigations of this magnitude in individuals who previously had mild to moderate SARS-CoV-2 infection are relevant for generating data in a population that is non-severely affected by the disease, and who may suffer from sequelae that are yet to be explored, with the intention to assist in the planning of rehabilitation programs.

As is known, age and metabolic disorders are factors related to poor prognostic of the COVID-19 disease. Our idea will be to study a sub-population that exhibits low risk of severe COVID-19 disease, and that in general will be not admitted to the hospital in intensive care. However, the consequences of infection during the long term were never studied.

Our main objective is to evaluate if physical activity levels and the number of adipose tissue depots are determinants of systemic inflammatory response and general clinical functions in adults who previously suffered from mild to moderate COVID-19, before and after specific vaccination, using a cohort prospective observational study. Additionally, we will explore modulatory effects of physical activity on immune responses to the virus, focusing on the function and metabolism of peripheral blood mononuclear cells (PBMCs) after in vitro exposure to homologous serum and/or recombinant coronavirus.

## 2. Methods

The protocol is registered with the Brazilian Registry of Clinical Trials (accessed date—21 September 2021 https://ensaiosclinicos.gov.br/). Any relevant changes will be updated on the site.

### 2.1. Study Design and Setting

A study design summary displaying the steps and evaluations of potential participants over approximately 18 months of follow-up is provided in Figure 1.

A follow-up period of approximately 18 months will be implemented. The procedures will be conducted for all participants at baseline, 6 weeks after vaccination, and 12 months after vaccination. After approval by the local Research Ethics Committee, patients previously infected with COVID-19 will be contacted to participate in the study through the local media and via electronic access to the participants’ database of Municipal Health Secretariat (MHS) of Presidente Prudente—São Paulo. Male and female patients aged 20–40 years with mild or moderate clinical COVID-19 infection will be recruited (intensive care cases will be excluded). An age-matched healthy control group that is negative for SARS-CoV-2 will also be enrolled. To detect previous SARS-CoV-2 infection, a lateral flow test for IgM and IgG antibodies will be conducted using internal anti-SARS-CoV-2 IgG and IgM ELISA kits. Participant exclusion criteria will include (1) presence of chronic non-communicable diseases, (2) smokers, (3) history of drug use, (4) medications such as anti-inflammatory drugs or antibiotics, (5) and frequent alcohol consumption (more than three times a week.

At each visit, body composition will be assessed using Dual X-ray Absorptiometry (DXA), bioimpedance, and ultrasonography. Accelerometers will be used to assess habitual physical activity. The cardiovascular function will be evaluated using an electrocardiogram, Doppler echocardiogram, and heart rate variability. The pulmonary function will be evaluated by spirometry. Peripheral and respiratory muscle strength will be measured using an electronic dynamometer and manovacuometer. Functional exercise capacity will be assessed using the 6-min walk test (6MWT). Mental health will be evaluated using questionnaires. Venous blood samples will be collected for immune phenotypic characterization and biochemistry assays (inflammatory and metabolic parameters). A protocol for ex vivo stimulation of whole blood with LPS will be used to imitate an inflammatory environment for determining the level of cytokines. Serum markers and markers of early cardiovascular function will be correlated with body fat and physical activity levels. Flow cytometry will be conducted and monocytes, T cells, and Natural Killer cell subsets in whole blood will be quantified and characterized. Additionally, after characterization and correlation of the determinants of SARS-CoV-2 severity with the amount of body fat and physical activity level and based on the hypothesis that individuals with low body fat and high physical activity levels have a better clinical outcome, we will examine immune cells from well-trained subjects (higher physical activity level and low-fat body) using an in vitro experimental model and compare them with age-matched non-trained subjects (control model).

Considering the effects of intermittent and regular physical activity on the immune system that can potentially mitigate viral infections, the molecular and biochemical mechanisms of action of immune cells in response to coronavirus will be detailed. Peripheral blood mononuclear cells (PBMCs) from well-trained participants will be collected and treated with homologous serum (collected from patients before and after vaccination) and recombinant coronavirus (inactive virus). After culturing, cytokine production will be determined by Luminex. PBMCs will be cultured for 24 h and then examined for phagocytosis, and cytokine and H_2_O_2_ production. Expression of *ACE-2*, *AT1R*, *AT2R*, *MASR*, *TLR-3*, *TLR-4*, *NOS*, *CETP*, *L-CAT*, and *ABCA1* genes will be analyzed using qRT-PCR. Energy metabolism will be characterized in terms of mitochondrial oxygen consumption rates (basal respiration, ATP production-linked respiration, maximal respiration, space respiratory capacity, proton leak-related respiration, non-mitochondrial respiration, bioenergetic health index, and coupling indexes) using the Seahorse XFe96 Extracellular Flux Analyzer (Agilent, Santa Clara, CA, USA). Mitochondrial DNA and transcripts of relevant oxidative phosphorylation (nuclear and mitochondria-encoded subunits of the ATP synthase and the mitochondrial respiratory chain, adenine nucleotide translocator, and voltage-dependent anion channel) will be obtained using qRT-PCR; antioxidant enzyme (superoxide dismutase I and II, glutathione peroxidase, glutathione reductase by colorimetric methods) activity and reduced and oxidized glutathione (GASH/GSSG) will be analyzed using high-performance liquid chromatography.

All reports generated from this study will follow the Strengthening the Reporting of Observational Studies in Epidemiology principles [29].

### 2.2. Assessment Schedules

General anamnesis, ergospirometry, body composition, physical activity level, and dietary intake measurements.

A general anamnesis will be taken, including sociodemographic characteristics, lifestyle habits, and self-rated health and medical history (family comorbidities, cardiopulmonary symptoms, and medication use). The questions in each medical domain are designed to allow self-rated assessments of pre-COVID-19 symptoms; symptoms that emerged during acute COVID-19; and persistent symptoms. The interview will also include the Medical Research Council (MRC) Dyspnoea Scale [30], the short form of the International Physical Exercise Questionnaire [31], and COVID-19 quality of life score by Post-COVID-19 Functional Status Scale [32].

All subjects will complete a peak oxygen uptake (VO2peak) protocol. Oxygen uptake will be measured with an incremental protocol on a friction-loaded cycle ergometer (Lode Excalibur, Groningen, The Netherlands) with expired gases analyzed using a breath-by-breath automated gas-analysis system (Quark CPET Cosmed, Rome, Italy). Stature will be measured on a fixed stadiometer (Sanny, São Paulo, Brazil) with an accuracy of 0.1 cm and a length of 2.20 m. Bodyweight will be measured using an electronic scale (FilizolaPL50, Filizola Ltd.a., São Paulo, Brazil) with a precision of 0.1 kg.

Body composition (total fat mass, trunk fat mass, abdominal fat, and lean mass) will be analyzed by ultrasonography using a 3.5-MHz probe located 1 cm from the umbilicus. Two ultrasonography measurements of visceral adipose tissue will be defined as the distance between the internal face of the same muscle and the anterior wall of the aorta. All measurements will be expressed in absolute and percentage values. Participants will be grouped into those with low-visceral adipose tissue (values below 3 ± 1 cm) and those with high-visceral adipose tissue (values above 5 ± 1 cm). Also, body composition (body mass, fat mass, musculoskeletal, bone, and fat-free mass from upper and lower limbs and trunk) will be analyzed using DXA (model DPX-MD, software 4.7, General Electric Healthcare, Lunar DPX-NT; Little Chalfont, Buckinghamshire, United Kingdom). Impedance measurements will be taken using a phase-sensitive bioimpedance analyzer (Bia Vitality, Harrisville, NH, USA) at a frequency of 50 kHz. Bioimpedance parameters [Resistance (R) and reactance (Xc)] were analyzed based on the BIVA procedures [33]. Phase angle will be calculated as the arctangent of Xc/R*180°/π. Total, intra- and extracellular water, body cell, and muscle mass will be estimated using specific bioimpedance-derived equations [34,35,36].

Nutritional information will be recorded at the initial screening from three-day food diaries that will be filled over two weekdays and one weekend day to better reflect typical intakes. Participants will meet with a nutritionist before testing and they will be instructed on how to complete the food diaries. Total energy, protein, carbohydrates, and fat intake will be analyzed using NutWinsoftware, version 1.5.

Physical activity levels will be measured using a triaxial accelerometer (GT3X+; ActiGraph, LLC, Pensacola, FL, USA) which gives valid estimates of physical activity in controlled and free-living environments. Participants will use an accelerometer worn above the waist for seven consecutive days during waking hours. A minimum of four days with at least 10 h/day will be defined as valid accelerometer data. Participants will be instructed not to use the accelerometer while bathing or performing water activities. Each day, participants will be responsible for recording when they went to sleep, when they woke up, and whether or not they took off the device. The data will be processed using the ActLife software version 6.9.2 (Pensacola, FL, USA).

### 2.3. Blood Collection and Analysis

At each time-moment, participants will visit the laboratory after overnight fasting. Forty milliliters of blood from an antecubital vein will be collected in a tube containing EDTA. To detect previous SARS-CoV-2 infection, a lateral flow test for IgM and IgG antibodies will be performed using internal anti-SARSr-CoV IgG and IgM ELISA kits. All patients (infected and non-infected with SARS-CoV-2) will be evaluated before (without vaccine) and after (with vaccine) the vaccination program. We plan to follow up the calendar vaccination of the São Paulo Government and recruit patients according to this schedule. Please, access information about the São Paulo State vaccination calendar in the link (https://www.prefeitura.sp.gov.br/cidade/secretarias/saude/vigilancia_em_saude/doencas_e_agravos/coronavirus/index.php?) (accessed on 5 January 2021). Phenotypes of immune cells will be characterized using multicolor flow cytometry, and monocytes, T cells, and Natural Killer (NK) cell subsets in whole blood will be quantified and characterized. T lymphocytes will be identified based on their positivity for CD3 and typical light scatter. Among positive CD3 cells, CD4^+^ T and CD8^+^ T cell subsets will be quantified. Based on the expression of CD14 and CD16, the classical (CD14++CD16−), intermediate (CD14++CD16+), and non-classical (CD14^+^CD16^++^) monocytes will be quantified. The NK cells will be defined as CD3^−^CD56^+^, and the two major subsets of NK cells, CD56^hi^CD16^−^ (CD56^bright^) and CD56^lo^CD16^+^ (CD56^dim^), will be identified. Biochemistry assays, including measurements of non-ester fatty acid (NEFA), glucose, triacylglycerol, total cholesterol, and HDL-c will be analyzed using a colorimetric method. Cortisol, insulin, and apolipoprotein A1 concentrations will be analyzed using ELISA. To determine the inflammatory status of the participants, the plasma concentrations of IL-1β, IFN-α, IFNγ, TNF-α, MCP-1, MIP-1β, IL-8, IL-10, IL-12p70, IL-17A, IL-18, IL-23, IL-33, IL-1ra, IL-2, IL-6 will be measured using the LUMINEX method.

An in-vitro analysis will be conducted to determine the mechanisms underlying the immune and inflammatory responses to the SARS-CoV-2 virus as a function of physical fitness. Well-trained and age-matched non-trained controls negative for SARS-CoV-2 will be recruited as detailed in the methods section. Phenotypic characteristics of monocytes, T, and NK cell subsets in blood samples will be measured as described above. Whole blood will be stimulated with recombinant human coronavirus to determine the cytokine profile. PBMCs from well-trained participants will be collected and treated with homologous serum (from the main study collected both before and after vaccination) and recombinant coronavirus (inactive virus). PBMCs will be examined after a 24 h culture period to measure phagocytosis and production of cytokines and H_2_O_2_. *ACE-2*, *AT1R*, *AT2R*, *MASR*, *TLR-3*, *TLR-4*, *NOS*, *CETP*, *L-CAT*, and *ABCA1* gene expression will be analyzed using qRT-PCR and metabolic analysis will be conducted using Respirometry (Seahorse).

The following sub-tasks will be implemented, as described:

**T1**: Stimulation with recombinant coronavirus: Whole blood samples from each participant will be divided into two tubes. Ten milliliters will be incubated with 100 µL of PBS and, 10 mL will be incubated with recombinant human coronavirus 229E (MyBiosource^®^) [29]. Samples combined with either PBS or human coronavirus 229E will be agitated for 120 min at room temperature. After this, the serum will be extracted by centrifugation at 690× *g* for 15 min at 4 °C, aliquoted, and stored at −80 °C for later use. Concentrations of IL-1ß, IL-2, IL-4, IL-6, IL-8, IL-10, IL-17, INF-γ, and TNF-α will be determined in non-stimulated (PBS) and stimulated (coronavirus 229E) samples using the Luminex method.

**T2:** PBMC culture and Cytokine Production: 30 mL of heparinized blood will be collected from both groups (well-trained and non-trained). PBMCs will be isolated from whole blood using the density gradient technique (Ficoll, d: 1.077, Sigma). Cytokine production will be measured separately using 1 × 10^5^ cells, which will be incubated in 96 Well plates containing RPMI 1640 medium enriched with 2 mM glutamine, 50% homologous serum (from patients in the main study), or human coronavirus 229E or vehicle solution as previously described [29]. After 48 h of incubation, the concentrations of IL-1ß, IL-2, IL-4, IL-6, IL-8, IL-10, IL-17, INF-γ, TNF-α will be measured using the Luminex method.

**T3:** Analysis of Angiotensin-Converting Enzyme in PBMC: *ACE-2*, *AT1R*, *AT2R*, *MASR*, *TLR-3*, *TLR-4*, *NOS*, *CETP*, *L-CAT*, and *ABCA1* expression will be measured using qRT-PCR, as previously described [30].

**T4**: Metabolic Analysis in PBMCs using Respirometry (Seahorse): samples will be plated in a 96-well xFe96 plate (Agilent) at three different densities. The density that best responds to the test will be used for subsequent experiments. For extracellular acidification (ECAR) studies, the Glycolytic Stress assay will be used. The drugs, 2-deoxyglucose, oligomycin, and acute glucose, will be used. To measure the consumption of mitochondrial oxygen (OCR), the MitoStress assay will be performed using the drugs oligomycin, CCCP (Carbonyl cyanide m-chlorophenylhydrazone), rotenone, and antimycin. The concentrations of all drugs will be standardized for better trial performance. All analysis will be done using Seahorse XF96 Extracellular flux Analyzer.

### 2.4. Heart Function, Structure, and Heart Rate Variability (HRV)

Resting echocardiography will be performed on a Vivid S6 device (GE Healthcare) in accordance with the guidelines. The echocardiographic data analyzed will include: left ventricular (LV) ejection fraction, LV volume, and LV mass, regional or global wall motion abnormalities, LV diastolic function markers (mitral inflow with E/A ratio, tissue Doppler with lateral e′, and ′ septal, mean E/and ′), volumes and surfaces of the left and right atrium, right ventricular (RV) function markers (TAPSE, S wave ′, pulmonary artery systolic pressure (PAPs) estimated from the peak velocity of tricuspid regurgitation (RT)) will also be evaluated based on international recommendations [37]. Electrocardiogram (ECG) measurements will be performed at rest and in an exercise test on a treadmill (Inbrasport, Ergo13 for windows version 3.1.0.95) (Bruce protocol). Resting ECG will be measured using ECG Mac 800 (GE Healthcare), while exercise ECG will be measured using Edan SE-1515 DX12 (Edan, Shenzhen, China) with aerobic protocol performed on a treadmill. During the resting ECG, heart rate (HR), arrhythmias, conduction disturbances, and ventricular repolarization anomalies will be evaluated. During the exercise ECG, the maximum HR reached (absolute and as a percentage of the theoretical maximum HR for the age), rest/peak blood pressure, supraventricular and/or ventricular arrhythmias and/or T wave/ST segment anomalies and/or driving disorders will be recorded.

Heart rate will be recorded beat-to-beat using a cardio-frequency meter (Polar RS800CX, Polar Electro, Kempele, Finland) at a 1 kHz sampling rate to assess cardiac autonomic modulation. Using the chest strap and monitor, the individuals will be placed in a sitting position and remain at rest with spontaneous breathing for 25 min. The temperature of the room will be maintained at 20–23 °C and data will be collected in the morning, after a 4 h minimum fasting period. Participants will be asked to avoid consuming coffee, chocolate, or alcohol and take medications the night before the procedure. Heart rate variability will be performed on 256 consecutive RR intervals from the most stable segment of the tachogram. Only series with less than 5% error will be considered suitable for analysis. The data will be processed using the Polar Precision Performance software (Polar Electro, Finland) in moderate mode, followed by visual inspection. HRV will be assessed using linear methods (time and frequency domains and Poincaré Plot quantitative and qualitative analysis) using the Kubios^®^ HRV software (version 2.2, Kuopio, Finland).

### 2.5. Physical Functioning

Functional exercise capacity will be assessed using the 6-min walk test (6MWT), according to the guidelines of the European Respiratory Society/American Thoracic Society [38].

Handgrip strength will be estimated using an electronic dynamometer (Camry Electronic Handgrip Dynamometer, Model EH 101, Taiwan). Each test will be performed three times with a one-minute interval between the tests. The highest value obtained will be recorded for analysis. Peak force values will be measured.

### 2.6. Pulmonary Function and Respiratory Muscle Strength

Spirometry will be performed using a digital spirometer (MIR-Spirobank version 3.6) according to the guidelines for pulmonary function tests [39]. The results will be interpreted based on the recommendations of the American Thoracic Society and the European Respiratory Society [40] and the results compared with normative data from a Brazilian population [41]. The spirometric criteria for airflow limitation will be a fixed postbronchodilator ratio of FEV1/FVC < 0.70.

Respiratory muscle strength will be assessed using a manovacuometer (MVD 300, Globalmed, Porto Alegre, Brazil). The participant will be asked to sit in an upright position, with the chest and feet supported. Using a nose clip, the participant will be instructed to hold the manometer and press the mouthpiece firmly against the lips, avoiding air leakage. They will be asked to make a maximum inspiration of the residual volume to measure MIP and a maximal expiration of total lung capacity to determine MEP. In each evaluation, three measurements of MIP and MEP will be performed, with rest intervals of 30–60 s between measurements. The highest value will be recorded.

### 2.7. Mental Health

Mental health will be assessed employing questionnaires of cognitive function and anxiety and depression. The quality of sleep will also be evaluated. The Digit Span (numbers) questionnaire assesses the ability to focus and maintain attention, and the mental manipulation of information has two stages. Both conditions require participants to verbally repeat requested sequences in direct or reverse order. Both forward and reverse order has a total score of 14 points.

The Brazilian version of the beck depression inventory (BDI), containing 21 items, will be used to evaluate the intensity of depression symptoms, with responses rated on a Likert scale [42]. Scores range from 0 to 63 points (0–11 minimal; 12–19 mild; 20–35 moderate; and 36–63 severe). The Brazilian version of the Beck Anxiety Inventory (BAI), consisting of 21 statements, will be used to assess anxiety levels [42], with responses rated from 0 to 63 points (0–10: minimal anxiety; 11–19: mild anxiety; 20–30: moderate anxiety; and 31–63: severe anxiety). The HADS aims to measure symptoms of anxiety and depression and consists of 14 items, seven items for the anxiety subscale (HADS Anxiety) and seven for the depression subscale (HADS Depression). Each item is scored on a response scale with four alternatives ranging between 0 and 3. After adjusting for six items that are reversed scored, all responses are summed to obtain the two subscales. Recommended cut-off scores are <7 for improbable cases, 8–11 for doubtful cases, and ≥12 for probable cases [43].

The Epworth Sleepiness Scale (ESS), which assesses the nature and occurrence of daytime sleepiness, will be used to measure sleep quality [44]. It is self-administered and measures the possibility of napping in eight everyday situations. To grade the possibility of napping, the individual uses a scale from 0 to 3, where 0 is none and 3 is the highest probability of napping. Using a total score >10 as the cutoff point, it is possible to identify individuals with a high probability of EDS. Scores greater than 16 are indicative of severe sleepiness. The Pittsburgh Sleep Quality Index (PSIQ) questionnaire assesses sleep quality over the previous month [45]. It consists of 19 self-administered questions grouped into 7 components (subjective sleep quality, sleep latency, sleep duration, habitual sleep efficiency, sleep disorders, use of sleep medications, and daytime dysfunction) with weights distributed on a scale of 0 to 3. The scores of these components are then combined to produce an overall score ranging from 0 to 21, where the higher the score, the worse the sleep quality. An overall PSIQ score >5 indicates that the individual has great difficulties in at least 2 components or moderate difficulties in more than 3 components [46].

### 2.8. Data Capture and Management

Data from interviews, scales, and tests will be captured and stored in real-time using web-based google forms developed by our research team. The tabulation of results from manual tests will be performed by experienced research staff directly in the goggle drive database shared with the responsible research team. A team of experts will manage the database and they will provide access to the different research groups involved to enable them to conduct interim and final statistical analyses.

### 2.9. Summarization of Clinical Information and Feedback to Participants

The data gathered from follow-up assessments of participants previously infected with COVID-19 will be summarized in short health reports to be used for the benefit of long-COVID sufferers in need of clinical care. Potentially relevant clinical information will be fed back directly to the participant or included in a written report to be forwarded to the health provider who will continue to care for the individual.

### 2.10. Sample Size Estimation and Planning for Data Analysis

A Cohort of participants previously infected with SARS-CoV-2 and age-matched healthy participants will be randomly selected from the target population. Sample size calculation (G*Power, version 3.0; power = 0.80; α level = 0.05; medium-size effect) provided an estimated 75 participants. Therefore, 100 male and female participants aged 20–40 years will be recruited (over-recruitment compensates for dropouts during follow-up).

Participants will be stratified based on habitual physical activity levels (assessed by accelerometry-low versus normal physical activity levels), body composition (normal weight versus overweight or obese), and SARS-CoV-2 status (positive versus negative). In addition, a sub-analysis will be stratified based on sex (men vs. women).

## 3. Discussion

This protocol describes an ambitious follow-up study that will be conducted in adults previously infected with COVID-19 (before and after vaccination program), highlighting long-term consequences of COVID-19 and its correlation with physical activity, body composition, and cross-linked with vaccination in the Brazilian population. Our hypothesis links reduced visceral adipose tissue deposits (which exhibit an anti-inflammatory profile, as hypoleptinemia and hyperadiponectinemia) with higher physical activity levels as central factors in reducing the severity of SARS-CoV-2 aggression, promoting metabolic programming of immune cells (favoring anti-inflammatory response), protecting the cardiovascular system, and general clinical profile.

Patients with COVID-19 who were consistently inactive and malnutrition had a greater risk of hospitalization, admission to the ICU, and death due to COVID-19 compared with patients who were consistently meeting the minimum physical activity guidelines (150 min/week) [27,47,48]. Longitudinal cohort studies of individuals who developed severe COVID-19 are being conducted to investigate possible symptoms and long-term clinical and functional alterations [49,50]. Longitudinal follow-up studies reporting persistent symptoms and sequelae of COVID-19 in mild and moderate patients are also needed in order to understand the intrinsic physiological mechanisms involved in these processes, their potential protective factors, and public health policies needed for treatment in the post-pandemic period in order to substantially minimize long-term chronic sequelae.

In addition to investigating an overlooked population, one advantage of the study proposed here is that we will conduct comprehensive symptom and clinical assessments of long-COVID manifestations pre- and post-vaccination. Potential limitations include the fact that we will rule out the presence of current re-infection based only on the absence of clinical signs and symptoms, rather than on the basis of a negative RT-PCR test. We will also only assess a specific, young population.

In summary, studies seeking solutions to problems associated with the long-term consequences of COVID-19 are of great interest worldwide. Since this follow-up study will be conducted over several months, preliminary data analysis may encourage our team and other researchers to conduct interventional clinical trials targeting specific long-term manifestations of COVID-19. Finally, we hope that the clinical information fed back to the participants involved in this research will facilitate their access to healthcare.

### Limitations

In cases of the COVID-19 re-infection, this condition will be reported, and we will put major attention to the results. The patient will be considered an outlier and may be removed from the analysis.

## 4. Conclusions

This study will allow observing the possible role of physical fitness to mitigate the deleterious effect of mild to moderate COVID-19 infection from physiological to molecular level. This approach will permit a better knowledge about the surviving mechanisms, given by the immunomodulatory role of physical fitness.

## Figures and Tables

**Figure 1 ijerph-18-13249-f001:**
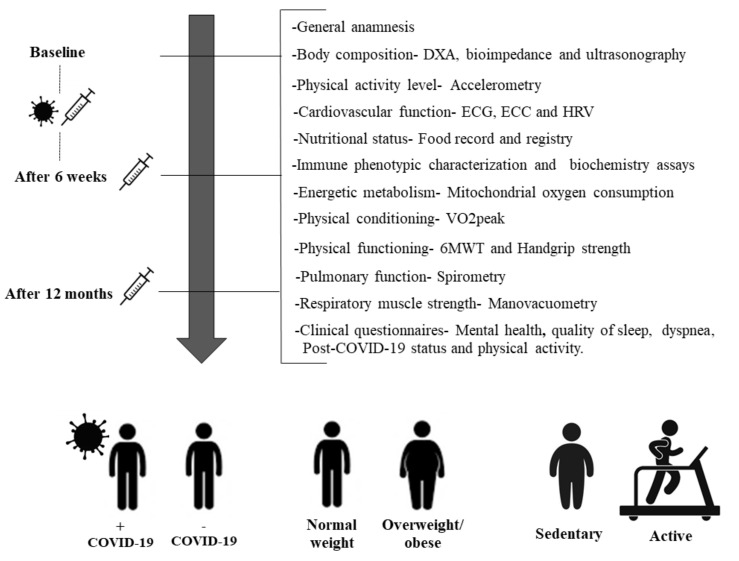
Study design. Steps and evaluations of potential participants over the follow-up period. DXA: Dual X-ray Absorptiometry; ECG: Electrocardiogram; ECC: echocardiography; HRV: heart rate variability; VO2peak: Peak oxygen uptake; 6MWT: 6-min walk test.

## Data Availability

No new data were created or analyzed in this study. Data sharing is not applicable to this article.

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
