# Peer review of "Modulatory Effects of Physical Activity Levels on Immune Responses and General Clinical Functions in Adult Patients with Mild to Moderate SARS-CoV-2 Infections—A Protocol for an Observational Prospective Follow-Up Investigation: Fit-COVID-19 Study"

_ijerph, 2021, doi:10.3390/ijerph182413249_

Round 1
Reviewer 1 Report
Dear authors,
Thank you very much for the opportunity to evaluate your script. Undoubtedly, this is a very important topic because our knowledge is very little about the symptoms of long-COVID-19. The research shows enormous potential and requires enormous financial resources and human work. Before the final approval decision, I have a few doubts, which are listed below:
1. There are inconsistencies in the methodological description, first the authors suggest (in the introduction) that the observation period will begin 6 weeks after acute COVID-19 and then 12 months after vaccination. While in the methodology section they indicate 6 weeks after vaccination and 12 months after vaccination - please standardize the description.
2. As part of the introduction, I would add information about the impact of nutritional status on the risk of death in the course of COVID-19, e.g. one of the most recent publications (Czapla, M .; Karniej, P; Juárez-Vela, R; Łokienia, K. The Association between Nutritional Status and In-Hospital Mortality among Patients with Acute Coronary Syndrome — A Result of the Retrospective Nutritional Status Heart Study (NSHS). Nutrients 2020, 12, 3091. https://doi.org/10.3390/nu12103091)
3. Will there be vaccinated people in the control group? Unvaccinated? Before vaccination? Will a vaccination declaration be a condition for participation in the study? What in the case of people who have been infected with COVID-19 and have not developed antibodies, or will the level of antibodies fall below normal during the measurement period? What idea do the authors have for this thread? Will there be verification tests, e.g. cellular response after disease?
4. Why was it decided to choose the age group 20-40?
5. In addition, what are the plans for patients who will fall ill with COVID-19 again during the study?
6. What is the estimated cost of the entire project? Was funding for the study successful?
Reviewer 2 Report
Dear Authors, I have read your manuscript with interest.
The current manuscript titled: "Modulatory Effects of Physical Activity Levels on Immune Responses and General Clinical Functions in Adult Patients with Mild to Moderate Covid-19 Infections – A Protocol for an Observational Prospective Follow-up Investigation: Fit-Covid 5 Study" represents an important analysis of evolving field of Infectious Diseases and Internal Medicine.
In my opinion, these are the adjustments which should be made to increase the value of your manuscript:
- Line 4-6: I suggest Authors to change “Covid-19 Infections” to “SARS-CoV-2 Infections” and “Fit-Covid Study” to “Fit-COVID-19 Study”.
- Please, write the latin words with italic font (i.g., versus, in viro, etc.).
- Line 65: I suggest Authors to rename “Discussion” section to “Conclusions”.
- Line 68: “Which”, write please without capital letter.
- Line 70: “Do”, write please without capital letter.
- Line 80: To common symptoms, add please “anosmia, ageusia”.
- Line 83-84: In sentence “…patients with mild to moderate COVID-19 who are often overlooked.”, after COVID-19, add please “form”.
- After the Figure 1 title, please add the abbreviations explications that are mentioned in the figure.
- In Figure 1, change please “dispnea” to “dyspnea”.
- Line 169-170: Please, explain why the abbreviation for “Municipal Health Secretariat” is (SMS) and for “Presidente Prudente” is SP?
- Line 170-171: On what basis did you establish the inclusion criterion of patients aged between 20 and 40 years? Please explain your decision in the text.
- Line 176: Why do you want to exclude smoking patients from the study?
- Will you use quality of life scores?
- Add please your future study limitations.
- The References does not meet the Journal requirements, please adapt the articles list according to the recommendations for authors.
- The manuscript contains many punctuation errors, please revise the text (lines 99, 128, 130, 132, 139, 273, etc.).
Good luck!
Reviewer 3 Report
This was an interesting study.
There is only a minor error in the manuscript where you mentioned on page 4 line 166 that the first follow up was after the vaccination but from the rest of the manuscript, I understand that the first follow up was 6 weeks after acute COVID-19 infection:
The procedures will be conducted for all participants at baseline, 6 weeks after vaccination, and 12 months after vaccination.
Also could you please clarify if by DXA you mean Dual X-ray Absorptiometry.
Reviewer 4 Report
Summary of the research and overall impression
The authors present a study protocol for a prospective observational cohort study to assess the influence of physical fitness/activity levels on the severity of SARS-CoV-2 infection and long-COVID.
The protocol examines a novel aspect of physical characteristics that may influence the progression and clinical course of long-COVID. The description of the protocol is well-executed and contains all relevant aspects necessary to conduct the study. The main weakness of the study is the lacking distinction between physical activity and fitness, the focus on particular immune cell subtypes only, and the lacking distinction between the assessment of male and female patients.
Overall, this protocol evaluates a potentially important aspect of long-COVID that warrants to be considered for publication.
Specific areas of improvement
-Major issues
1) One major drawback of the study is the exclusion of severe infections, as the influence of physical fitness on the occurrence of severe infections would be interesting to examine – the highest effects are to be expected there. Can you please elaborate on the reasoning as to why these patients would be excluded?
2) The expressions physical fitness and activity levels seem to have been used interchangeably in the manuscript. In particular, you appear to want to measure fitness, but only the general activity levels were assessed. Although connected and certainly correlated, a distinction between the two should be made. For example, a lean person, who walks for many hours a day does not necessarily have to be more physically fit than someone who conducts an hour of high-impact exercise once a week.
3) No differences between men and women are made in the assessment of the body fat distance ratio measurement. Fat deposits in women after birth could have a confounding effect on the measurement and indicate that women are generally less physically fit than men using your measurement.
4) Please elaborate on how the term “well-trained” is characterized in particular and what the cut-off lines are.
3) Line 131: The authors aim to demonstrate that this will alleviate the severity of the pandemic, which also includes economical and societal consequences; however, the data only supports that it may be relevant toward the consequences of SARS-Cov-2 infections in patients.
4) Is there a non-vaccinated control group? Is this control group being tested for the presence of a SARS-Cov-2 infection?
5) Please elaborate in more detail in your introduction, why the study focusses on the investigation of innate immune cell responses, for instance the effect that innate immune response cytokine profiles may have on the prediction of the clinical course.
6) Figure 1: The list items are not clearly allocated to the different timepoints, i.e., baseline, 6 weeks, and after 12 months. Please revise this accordingly.
7) Based on my previous comment (5), I would suggest to clarify again, why the cytokine expression was only examined in PBMCs, as cytokines are produced by a range of immune cells, including cells of the adaptive immune system.
8) Line 205: Culturing PBMCs for 48 hours may lead to apoptosis in some cells, such as neutrophils. This may interfere with the expression of cytokine generation. Will the expressed cytokines produced during this timecourse be stable for that long a time period?
9) Please elaborate, why these genes are important: ACE-2, AT1R, AT2R, MASR, TLR-3, TLR-4, NOS, CETP, L-CAT, and ABCA1 genes (line 206).
Minor issues
1) Please ensure that all appropriate scientific terms are formatted in italics. Example: In-vitro (Line 57).
2) Please clarify here, whether the blood used was from patients or healthy volunteers (line 58).
3) Please adapt the style of the list to show a consistent use of sentence case, or lower case at the beginning of a sentence, and a consistent style of punctuation, i.e., questions should end with a question mark (Line 66+).
4) Line 99: Space missing between diseases and (4).
5) Line 223: This sentence does not appear to be complete.
6) Line 269: The term “time-moment” is unclear from the context. If a general expression should be used, the term “time-point” may be a suitable alternative?
7) Line 290: Why are neutrophils not included in the characterization, given their important role in Covid infections?
8) Line 292: Whole blood of who? What is the rationale of stimulating the blood of patients previously infected with the virus?
9) No information on the collection procedure to obtain informed consent was provided in the protocol.
10) Line 463: Why is no test performed? As the clinical signs and symptoms in previously infected individuals may be reduced, this is an important point to consider.
11) Given the lacking accuracy and reliability of IgG/IgM measurements several months after viral exposure, how can you reliably determine that at the baseline, people are not suffering from long-COVID?
12) Will the scientists conducting the assessments be blinded?
13) Please elaborate the data security of personal data, if these are to be stored in a google database (is the patients data anonymized, do the researchers have access to this information?)
14) Please ensure that a consistent font and font size is used throughout the manuscript (Example: line 480).
Round 2
Reviewer 1 Report
Thank you very much for responding to my concerns.
After the introduced corrections, the quality of the article increased significantly, however, in my opinion, most of the answers should be included in the text, especially questions 3 4 and 5
Author Response
The authors appreciated the comment. The information was inserted in text.
Reviewer 2 Report
I agree with the new submission.
Author Response
The authors appreciated the comment.